# Evaluation and Characterization of the Insecticidal Activity and Synergistic Effects of Different GroEL Proteins from Bacteria Associated with Entomopathogenic Nematodes on *Galleria mellonella*

**DOI:** 10.3390/toxins15110623

**Published:** 2023-10-24

**Authors:** Abraham Rivera-Ramírez, Rosalba Salgado-Morales, Janette Onofre-Lemus, Blanca I. García-Gómez, Humberto Lanz-Mendoza, Edgar Dantán-González

**Affiliations:** 1Center for Population Health Research, National Institute of Public Health, Cuernavaca 62100, Mexico; abraham.rivera@uaem.mx; 2Biotechnology Research Center, Autonomous University of the State of Morelos, Av. Universidad 1001, Chamilpa, Cuernavaca 62209, Mexico; rosalba.salgadomo@uaem.edu.mx (R.S.-M.); janette.onofre@uaem.mx (J.O.-L.); 3Biotechnology Institute, National Autonomous University of Mexico, A.P. 510-3, Cuernavaca 62250, Mexico; blanca.garcia@ibt.unam.mx; 4Center for Research on Infectious Diseases, National Institute of Public Health, Cuernavaca 62100, Mexico; humberto@insp.mx

**Keywords:** chaperonin, GroEL, insecticidal, *G. mellonella*, synergism

## Abstract

GroEL is a chaperonin that helps other proteins fold correctly. However, alternative activities, such as acting as an insect toxin, have also been discovered. This work evaluates the chaperonin and insecticidal activity of different GroEL proteins from entomopathogenic nematodes on *G. mellonella*. The ability to synergize with the ExoA toxin of *Pseudomonas aeruginosa* was also investigated. The GroELXn protein showed the highest insecticidal activity among the different GroELs. In addition, it was able to significantly activate the phenoloxidase system of the target insects. This could tell us about the mechanism by which it exerts its toxicity on insects. GroEL proteins can enhance the toxic activity of the ExoA toxin, which could be related to its chaperonin activity. However, there is a significant difference in the synergistic effect that is more related to its alternative activity as an insecticidal toxin.

## 1. Introduction

GroEL chaperonin is an ATP-dependent molecular chaperone that is universally present in all life forms and is one of the most conserved proteins in living organisms [1]. GroEL is composed of 14 identical 57 kDa subunits that form two homoheptameric rings (each with a large central cavity) that are stacked back to back with a molecular weight of 800–900 kDa. Structurally, GroEL presents three functionally different domains: apical, intermediate, and equatorial [2,3]. Each of these domains carries out essential functions for the activity of this protein. The apical domain (residues 189–377) is rich in hydrophobic residues and is responsible for the recognition of GroES and unfolded protein substrates [4,5]. The intermediate domain, on the other hand, was found to form a highly dynamic structure that has the function of a hinge, extending over two regions within the polypeptide chain and covering residues 134 to 190 and 377 to 408, respectively. This domain is responsible for transmitting signals between the apical and equatorial domains due to the binding of ATP and protein substrates that allosterically regulate the conformational states of the complex [6]. The equatorial domain, on the other hand, covers the two ends of the polypeptide chain, with residues 1 to 133 and 409 to 523 being part of its structure; this is in charge of carrying out the ATPase activity, as well as inter- and intra-subunit interactions within the double-ring complex [7]. Their primary function is to help other proteins fold correctly in the cell. The GroEL–GroES system of *E. coli* has been the most extensively studied [8,9].

However, apart from this primary task, additional so-called “moonlighting” functions of GroEL proteins unrelated to their folding activity have been reported in recent years [10,11]. Many of these activities reported for GroEL have been discovered in microorganisms that establish symbiotic relationships with their hosts. Perhaps one of the most relevant activities found for the GroEL protein is its action as a toxin against insects, which is why it has been considered an insecticidal factor in some bacteria [12,13]. Yoshida et al. [14] reported an insecticidal protein factor from *Enterobacter aerogenes*, an endosymbiont present in the saliva of *Myrmeleon bore* ant larvae, which rapidly paralyzed and killed cockroaches of the genus *Blatella germanica* when injected into the hemocoel. Sequence analysis showed a highly conserved identity to the structural homolog of GroEL in *E. coli*. Also, several amino acid residues of *Enterobacter* GroEL were found to be involved in toxicity (Val 100, Asn 101, Asp 338, Ala 471), and substitution of these amino acids in the *E. coli* GroEL protein conferred toxicity to the protein. So far, nothing is known about the mechanism underlying the toxicity; however, neither GroES nor ATP binding was necessary for GroEL toxicity. Subsequently, Khandelwal et al. [15] found, a bacterial symbiont of entomopathogenic nematodes of the genus *Steinernema* in culture broths of *Xenorhabdus nematophila;* the outer membrane vesicles (OMVs) contained insecticidal factors, which included the GroEL protein (GroELXn) as a significant constituent. Again, co-chaperonin GroESXn was not found in OMVs, suggesting a specific export pathway for GroELXn. Joshi et al. [16] evaluated purified GroELXn protein in neonate larvae of *Helicoverpa armigera,* showing insecticidal activity when administered orally in an artificial diet. In addition, the GroELXn protein could bind to brush border membrane vesicles derived from the larval gut, suggesting that the insect gut’s peritrophic chitinous membrane may be a primary target for GroELXn binding. Mutation analysis revealed that two amino acids of GroELXn (Thr347 and Ser356) were critical for the binding and toxicity of GroELXn. Also, it was reported that the toxicity of GroELXn is entirely unlinked to its folding activity since it does not require the co-chaperonin GroESXn. Likewise, two other reports of GroEL homologs of *Xenorhabdus budapestensis* and *Xenorhabdus ehlersii* showed insecticidal activity when evaluated through injection into the hemocoel of *Galleria mellonella* larvae. Thus, in these entomopathogenic bacteria, GroEL has evolved as an essential virulence factor [17,18].

In this work, we evaluated the insecticidal activity of different recombinant GroEL proteins from nematode-associated bacteria to determine if the activity has evolved in other bacterial members in this particular entomopathogenic nematode–bacteria interaction. Interestingly, it has been observed that some chaperones can cooperatively enhance the insecticidal activity of Cry1A toxin from *Bacillus thuringiensis* [19,20]. Therefore, we found it interesting to evaluate the synergistic effect of GroEL with exotoxin A (ExoA) of *Pseudomonas aeruginosa* NA04 isolated from entomopathogenic nematodes.

## 2. Results

### 2.1. Evaluation of the Insecticidal Activity of GroEL Proteins

Recombinant GroEL proteins were obtained from the pET28a vector construct and the amplified fragment corresponding to the *groEL* gene for each strain using specific primers; see Table 1 (*Alcaligenes faecalis* MOR02, GroELAf; *E. coli* DH5α, GroELEc; *Pseudomonas aeruginosa* NA04, GroELPa; *Photorhabdus luminescens* HIM3, GroELPl and *X. nematophila* SC 0516, GroELXn). They were purified to homogeneity using a nickel-nitrilotriacetic acid column (shown in Appendix A). All of the proteins evaluated showed chaperonin activity in vitro, as they were able to refold the enzyme lactate dehydrogenase after it had been denatured by heat shock. No differences were observed in the refolding activity curves (shown in Appendix A).

The biological insecticidal activity of the proteins was evaluated through the direct injection of the protein into the insect haemocoel to simulate the secretion of the protein by the bacteria inside an insect when infected by an entomopathogenic nematode.

The GroEL protein showing the highest activity was GroELXn, with a significant difference (F = 8.932, Dfn = 4, DFd = 65; *p* < 0.0001). However, the activity of GroElEc also showed statistical significance when compared to the activity of the other GroEL proteins (Figure 1). The LD_50_ for GroELXn was 102.34 ng/larvae, while for the GroELEc protein, it was 895.51 ng/larvae, and the LD_50_ of the other proteins could not be determined due to their relatively low insecticidal activity. Yang et al. [18] reported the activity of a GroEL from *X. budapestensis,* which was also administered through injection into *G. mellonella* larvae. This protein also showed a dose-dependent effect, reaching an LD_50_ of 206.81 ng/larvae at 48 h. Likewise, Shi et al. [17] also reported the toxic activity of another GroEL protein from *X. ehlersii*. However, the LD_50_ for the latter was 760 ng/larvae. All proteins showed an activity that was considered basal at about 30% for the GroELAf, GroELPl, and GroELPa proteins at 48 h of treatment.

### 2.2. Evaluation of the Phenoloxidase Activity

A quite evident phenomenon was a change in the phenotype of larvae when injected with increasing doses of GroELXn protein, thus triggering a generalized blackening of the larval body after 15 min post-injection. Therefore, we decided to evaluate the activity of the phenoloxidase (PO) enzyme. This enzyme is a crucial component of the innate defense system in invertebrates and leads to the production of melanin compounds upon tissue damage or pathogen infection. This critical process is controlled by the enzyme phenoloxidase, which, in turn, is regulated in a highly elaborate manner to avoid the unnecessary production of highly toxic and reactive compounds [23,24].

**Figure 1 toxins-15-00623-f001:**
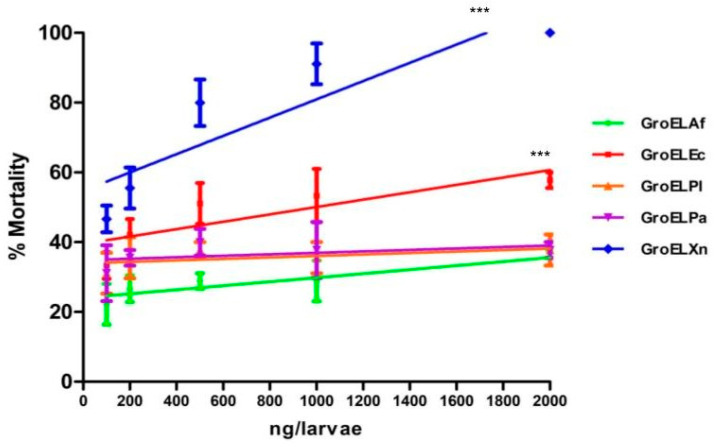
Evaluation of the insecticidal activity of GroEL proteins at 48 h post-injection. Individuals of *G. mellonella* were injected with the corresponding concentrations, which are indicated on the *x*-axis in a total volume of 10 μL. Each point on the graph represents the average of three individual experiments ± SD. The asterisks at the top indicate the degree of statistical significance (*p* < 0.0001) according to ANCOVA.

To carry out the experiment, we used the concentration corresponding to the LD_50_ of the GroELXn protein per larva. We also injected GroELAf, which showed the lowest insecticidal activity. Both proteins were used at the same concentration. As shown in Figure 2, the phenoloxidase activity was primarily increased in the treatment with the GroELXn protein. We observed a statistically significant increase when comparing the two enzymatic activity curves (F = 22.0069, DFn = 2, DFd = 39, *p* < 0.0001). Therefore, the results suggest that the mechanism by which GroELXn may be acting when injected into the insect hemocoel is through the overactivation of the prophenoloxidase system.

This phenomenon was also observed with GroEL from *X. budapestensis* (HIP57) and *X. ehlersii* (XeGroEL), which obscured the bodies of larvae treated with *G. mellonella* when injected at concentrations close to the LD_50_ values, which were 206.81 ng/larvae and 760 ng/larvae, respectively [17,18].

### 2.3. Protein Sequence and Structural Analysis

To investigate the possible sequence-level changes responsible for the disparity in activities among the different GroEL proteins, we performed multiple alignments with the primary sequence of each protein to observe the number of substitutions and the degrees of similarity between them. 

The proteins showed a percentage of similarity to GroELXn that showed the highest activity of 71.90% for GroELAf, 79.56% for GroELPa, 89.60% for GroELEc, and 93.61% for GroELPl. According to this analysis, the sequence with the highest degree of divergence was GroELAf, while it showed the closest phylogenetic relationship with its ortholog GroELPl. The analysis also showed that the different mutations between the proteins did not occur in a specific polypeptide chain region but were dispersed along the chain (Figure 3). The GroELXn protein had 155 various point substitutions compared to GroELAf; 88 of the 155 substitutions were in the equatorial domain, 47 were in the apical domain, and 20 were in the intermediate domain. In the case of the GroELPa protein, the protein with the second most significant divergence, there were 111 differences, of which 59 are in the equatorial domain, 36 were in the apical domain, and 16 were in the intermediate domain. The GroELEc protein had 57 point differences, of which 33 were in the equatorial domain, 18 were in the apical domain, and 6 were in the intermediate domain. The protein with the closest evolutionary relationship, GroELPl, had only 35 point substitutions with respect to GroELXn, 20 of which were in the equatorial domain, 13 of which were in the apical domain, and 2 of which were in the intermediate domain. Likewise, the GroELXn sequence presented 15 unique substitutions that could probably be involved in this protein’s more remarkable display of insecticidal activity compared with its homologs (Appendix A).

According to the reports on GroEL proteins that are toxic to insects, Yoshida et al. [5] mentioned the importance of four essential point mutations for the protein homolog coming from *E. aerogenes* to exert toxicity on cockroaches of the genus *B. germanica*; these were Val100, Asn101, Asp338, and Ala471. The analysis showed that the GroELXn protein had two mutations of the four that were indicated (Val100 and Ala471), so its structural homolog and this one could exert toxicity against insects through different mechanisms that do not involve the same amino acids. Also, Joshi et al. [7] reported the presence of two critical substitutions for toxicity in a homolog from *X. nematophila* (Thr347 and Ser356). These two residues were responsible for binding to intestinal epithelial glycoproteins in neonate *H. armigera* larvae, and when mutated with alanines, their toxicity was reduced by up to 80%. Our analysis indicated that the Thr347 residue was only present in the GroELXn sequence, while the S356 position was found in three protein sequences: GroELXn, GroELPl, and GroELAf.

### 2.4. GroEL Interacts Cooperatively with ExoA to Increase Toxicity

The synergistic activity of the toxins GroEL and exotoxin A was evaluated by combining both toxins in sublethal doses that produced mortality at rates between 20 and 30% in larvae of *G. mellonella*. For these experiments, we used only GroELXn, which showed the highest insecticidal activity, and GroELAf, which showed the lowest activity. Exotoxin A (50 ng) alone produced a mortality of 35.55%, while GroELXn (25 ng) produced 30% and GroELAf (25 ng) produced 2.22% 48 h after injection (shown in Figure 4). However, in the case of the mixtures of exotoxin A with both the GroELXn and GroELAf proteins, a cooperative effect on toxicity was shown when they were injected, resulting in a mortality of 100% for the first case (ExoA 50 ng/GroELXn 25 ng) and 68.88% for the second (ExoA 50 ng/GroELAf 25 ng) (Figure 4).

## 3. Discussion

In this study, we highlight two fundamental characteristics of GroEL proteins: their insecticidal activity and their synergistic effect on the activity of exotoxin A protein. These findings reinforce their recognition as moonlighting proteins. Interestingly, all GroEL proteins examined in this study exhibited insecticidal activity whether they originated from bacteria associated with entomopathogenic nematodes or from free-living bacteria, such as *E. coli.* Therefore, the ability of GroEL proteins to exert toxicity on insects is a general characteristic of theirs. However, we also observed a clearer adaptation in GroELXn to function as an insecticidal toxin, and that could be related to the unique substitutions it presents within its protein sequence.

Despite these proteins sharing a relatively high amino acid identity, their insecticidal activity varies significantly. This variation does not seem to be related to the percentage of similarity that they may share. For example, GroELAf exhibits the lowest activity and the highest divergence with GroELXn, which has the most increased insecticidal activity. However, notably, GroELPl has the highest percentage of similarity (93.61%) with GroELXn, but it exhibits low insecticidal activity that is similar to that presented by GroELAf. These data will be crucial for future research, particularly in elucidating the amino acids involved in insecticidal activity.

García-Gómez and collaborators reported the highly significant enhancement of the insecticidal activity of the Cry1A protein against a strain of *Plutella xylostella* NO-QAGE, which is resistant to the Cry1Ac toxin, when it was placed in the presence of the chaperones Hsp70 and Hsp90. In another work by the same group, García-Gómez et al. [10] compared the enhancement of the insecticidal activity of the Cry1Ac toxin against *P. xylostella* when it was mixed with Hsp70 from the target insect and GroEL from *A. faecalis*. PxHsp70 improved the toxicity of the Cry1Ab protoxin to *P. xylostella* larvae that were placed on an artificial diet in a dose-dependent manner, reaching almost 100% mortality with 250 ng/cm2 of Hsp70 (*p* < 0.0001). In contrast, the GroEL chaperone (250 ng/cm^2^) enhanced the insecticidal activity of the Cry1Ac protoxin by up to 40% (*p* < 0.0001).

In this work, we demonstrate for the first time that the enhancement of the toxic activity of the exotoxin A protein of *P. aeruginosa* is increased when a chaperonin such as GroEL is present. We also demonstrated that the potentiation phenomenon between both proteins is closely related to the intrinsic insecticidal activity of GroELXn.

## 4. Materials and Methods

### 4.1. Maintenance of Insects

*G. mellonella* was maintained at 22 ± 5 °C under controlled conditions with light/dark periods of 12:12 h and a relative humidity of 70 ± 10%. They were fed an artificial diet. Larvae in the final stage were selected and weighed for bioassay analysis.

### 4.2. Cloning, Expression, and Purification of GroEL Proteins and ExoA

Specific primers were designed for the PCR amplification of the *GroEL* sequences of *Alcaligenes faecalis* MOR02 (GenBank accession number KGP00134.1), *Photorhabdus luminescens* HIM3 (GenBank accession number OWO81362.1), *Pseudomonas aeruginosa* NA04 (GenBank accession number OWO88481.1), and *Xenorhabdus nematophila* SC 0516 (GenBank accession number MBA0017902.1), and the GroEL protein from *E. coli* DH5α was used as a control. In all cases, cut-off sites were added for the EcoRI and NdeI enzymes.

The coding sequences of the different GroEL proteins were cloned into the blunt vector pJET1.2 (Thermo Scientific, Vilnius, Lithuania), and the resulting constructs were subsequently transformed into *E. coli* DH5α cells following the protocol described by the manufacturer. Subsequently, the 1.7 kb fragments were ligated into the expression vector pET28a, and these constructs were transformed into *E. coli* BL21DE3, thus producing the recombinant strains GroELAf, GroELEc, GroELPa, GroELPl, and GroELXn. These cells were cultured at 37 °C and 150 rpm in Luria Bertani medium containing 50 μg/mL kanamycin. Once an optical density (OD600 nm) of 0.5 was reached, gene expression was induced by adding 1 mM IPTG for 4 h. The cells were washed with PBS 3 times and subsequently sonicated. The supernatant free of cell debris was filtered through an agarose–acetic acid–nickel–agarose affinity column (Ni-NTA Agarose QIAGEN, Hilden, Germany) at a temperature of 4 °C according to the protocol recommended by the manufacturer.

The cloning, expression, and purification of the ExoA protein (Genbank accession number OWO87729.1) without signal peptide were carried out with the same methodology as that described above.

### 4.3. Evaluation of the Insecticidal Activity of GroEL Proteins

To evaluate the insecticidal activity of the different GroEL proteins, last-instar larvae of *G. mellonella* were used. They were injected with increasing doses of protein using 100, 200, 500, 1000, and 2000 ng of protein per larva. Proteins were administered into the hemolymph of individual larvae through injection into the last anterior segment of the larval body with 0.3 mL 31 G × 6 mm Ultra Fine U-100 insulin syringes (BD Medical-Diabetes Care, Holdrege, NE, USA). A final volume of 10 µL was used for all treatments, and BSA (2000 ng) and Trizma buffer were used as negative controls. Three independent trials were carried out with 15 larvae per treatment. Mortality was recorded daily for three days.

The mortality curves were converted into semi-logarithmic linear regressions, and an ANCOVA was performed to detect significant differences between the different curves, with a rejection probability of *p* < 0.0001.

### 4.4. Sequence and Structure Analysis

The global alignment of the different GroEL sequences was performed with the Jalview application using the MAFFT program with the default parameters [25]. I-TASSER [26] was used to generate the three-dimensional structure, which was visualized and analyzed with Visual Molecular Dynamics (VMD) software (1.9.4a57; University of Illinois Urbana-Champaign, Urbana, IL, USA) [27,28].

### 4.5. Assessment of the Biochemical Activity of Chaperonin GroEL

Chaperonin activity was evaluated by using a refolding assay of the bacterial lactate dehydrogenase (LDH) enzyme (L3888, Sigma, Saint Louis, MO, USA) as described by Hristozova et al. [29]. The LDH enzyme was completely denatured by thermal shock at 56 °C for 60 min. As the reaction volumes were relatively small, even limited evaporation would change the reagent concentrations in the solution and compromise the measurements. Thus, we sealed the plates at the heating step. Cofactors and reporting agents were added just prior to avoid thermal decomposition. The reaction solution was prepared with 6.6 mM NADH (N8129, Sigma), 30 mM sodium pyruvate (P2256, Sigma), and 1 mM MgCl, 5 mM ATP (0220, VWR LifeScience) in a buffer with 200 mM Tris-HCl at pH 7.3. The enzyme concentration during the refolding reaction was 10 μM LDH at 25 °C. A five-fold excess of the molar concentration of GroEL with respect to the enzyme was added to the refolding reaction. The reaction mixture without protein, the mixture with LDH only (without GroEL), and the mixture plus LDH and GroEL (without ATP and without Mg^2+^) were used as controls. This assay was carried out in 96-well plates with a final volume of 200 μL for each treatment. In a microplate reader, the reaction was monitored every 10 min at 360 nm for 60 min at 25 °C.

### 4.6. In Vitro Phenoloxidase Activity Assay 

Last-instar *G. mellonella* larvae were used in this assay by injecting a dose of 100 ng GroEL or PBS into the hemocoel of each larva. Two hours after being freshly prepared, hemolymph (1:10 diluted in 100 mM Tris–HCl buffer, pH 6.5) was collected from a surface-disinfected caterpillar through an incision on the second proleg, which was the source of the enzyme. The reaction mixture was prepared by mixing the substrate of 0.1 mL 4 mg/mL DOPA (3,4-dihydroxy-L-phenylalanine, Sigma) with 1.3 mL 100 mM Tris–HCl buffer, pH 6.5, and 0.1 mL of hemolymph. The final mixture was continuously monitored for 60 min at a wavelength of 490 nm using an iMark™ Microplate Absorbance Reader (BioRad; Tokyo, Japan). ANCOVA was performed to detect significant differences between the two lines with a probability of rejection of *p* < 0.0001.

### 4.7. Determination of the Synergistic Effect of GroEL with Exotoxin A

The synergistic activity of GroEL and exotoxin A was evaluated by combining both toxins in sublethal doses to produce 10–20% mortality in larvae. The experimental units consisted of 10 sixth-instar larvae that were inoculated with exotoxin A at a concentration of 25 or 50 ng/larva, GroEL 25 ng/larva, or a mixture of both toxins at the concentrations described; the toxins were diluted in 10 mM Trizma buffer at pH 8.5. BSA at a concentration of 75 ng/larva and Trizma buffer were used as controls. Data are shown as the mean ± SD. The *p*-value was calculated by using Student’s *t*-test for two groups, and *p*-values < 0.05 were considered as statistically significant differences.

### 4.8. Statistical Analysis

Statistical analyses were performed with GraphPad Prism version 8.4.3 (GraphPad Software Inc., San Diego, CA, USA).

## Figures and Tables

**Figure 2 toxins-15-00623-f002:**
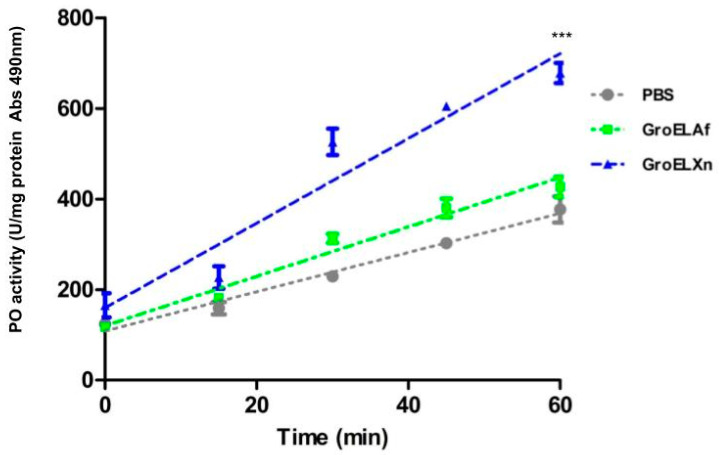
In vitro analysis of prophenoloxidase activity with *G. mellonella* hemolymph. The prophenoloxidase activity was measured by adding L-Dopa as an enzyme substrate, and the activity was monitored by using the absorbance at 515 nm. Points on the graph represent the average of 3 independent experiments ± SD. The asterisks indicate significant differences at *p* < 0.0001 when treatments were compared according to ANCOVA.

**Figure 3 toxins-15-00623-f003:**
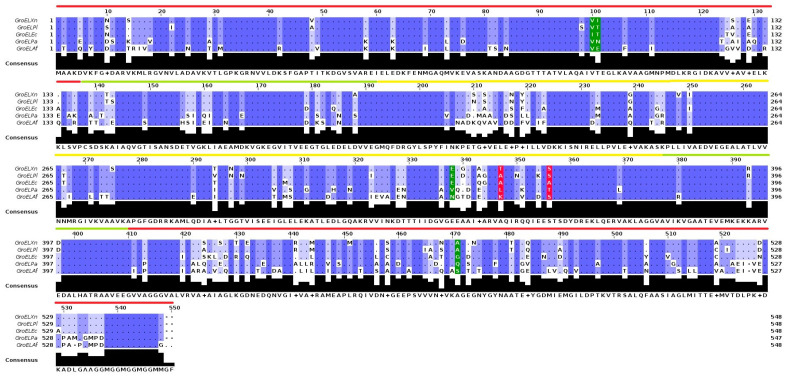
Structural analysis of GroEL proteins. Comparison of the protein sequences of the different GroEL proteins. Sequence alignment was carried out using the Jalview program and MAFFT with the default parameters, and the results were colored based on the percentage similarity. Colored in green are the key substitutions in conferring toxicity reported by Yoshida et al. for GroEL from *E. aerogenes.* Colored in red are the key substitutions in conferring toxicity reported by Joshi et al. for GroEL from *X. nematophila.* The red line at the top of the sequence corresponds to the equatorial domain (residues 2–136 and 410–525), the green line corresponds to the intermediate domain (residues 137–188 and 378–409), and the yellow line corresponds to the apical domain (residues 189–377).

**Figure 4 toxins-15-00623-f004:**
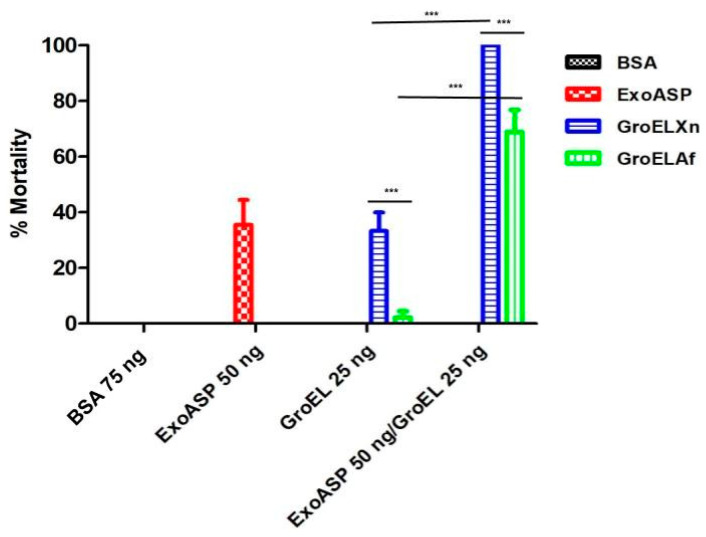
Analysis of the synergistic activity of GroEL proteins and exotoxin A. Percentage mortality of GroEL proteins (25 ng/larvae) against *G. mellonella* in the presence of exotoxin A protein (ExoA 50 ng/larvae). Fetal bovine serum (BSA 75 ng/larvae) was used as a control. Data with standard deviations represent the means of three treatments using 15 larvae per treatment in each repetition. Values of *p* < 0.05 indicated statistically significant differences: ***.

**Table 1 toxins-15-00623-t001:** Primers used for PCR amplification of the *groEL* and *exoA* genes coding for GroEL and ExoA proteins.

Name	Sequence (5′→3′)	Reference
*GroELAF_Fw*	CATATGACCGCAAAACAAGTTTACTTCG	[21]
*GroELAF_Rv*	GAATTCTTAGAAGCCGCCCATACCACCCATGC	
*GroELEc_Fw*	CCATATGGCAGCTAAAGACGTAAAATTCG	
*GroELEc_Rv*	GAATTCATTACATCATGCCGCCCATGCCAC	
*GroELPa_Fw*	CATATGGCTGCCAAAGAAGTTAAGTTC	In this work
*GroELPa_Rv*	GAATTCTTACATCATGCCGCCCATGC	
*GroELPl_Fw*	CATATGGCAGCTAAAGACGTAAAATTTGG	In this work
*GroELPl_Rv*	GAATTCTTACATCATGCCGCCCATACCG	
*GroELXn_Fw*	CATATGGCAGCTAAAGACGTAAAATTTG	[22]
*GroELXn_Rv*	GAATTCACATCATGCCGCCCATTCCAC	
*ExoA_Fw*	CATATGGCCGAGGAAGCCTTCGATCTCTCT	In this work
*ExoA_Rv*	GAATTCTTACTTCAGGTCCTCG	

## Data Availability

Not applicable.

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
