# Peer review of "Evaluation and Characterization of the Insecticidal Activity and Synergistic Effects of Different GroEL Proteins from Bacteria Associated with Entomopathogenic Nematodes on Galleria mellonella"

_toxins, 2023, doi:10.3390/toxins15110623_

Round 1

Reviewer 1 Report

Please confirm the attached file.

Author Response

Response to Reviewer 1

We appreciate the positive feedback from the reviewer. We have answered each of your points below.

  1.       L60-L69,Most GroELs are written as GroELXX. For example, GroEL of Alcaligenes faecalis is written as GroELAf, and GroEL of Pseudomonas aeruginosa is written as GroELPa. Similarly, most of the GroEL of Xenorhabdus nematophila is described as GroELXn, but from L60 to L69, it is described as XnGroEL. Shouldn't it be unified to GroELXn? Similarly, is XnGroES GroESXn?

Response:

We appreciate the observation; all questions have been satisfactorily answered. The names of the GroELs have been standardized, ensuring consistency throughout the manuscript.

  1. L71 Galleria mellonella mellonella

            L144  three individual ±DS →three individual ±SD

            L163  P>0.0001 →P<0.0001

 Response:

Thank you for these observations. The suggested correction has been made.   

3.       L174-185The explanations for L174 to L185 are expected to be related to Figure 3, but it is unclear which areas the "equatorial domain", "apical domain", and "intermediate domain" explained here refer to. Could you please explain a little more about each domain?  

Response:

Thank you for this excellent observation. We have made modifications to Figure 3. We have added lines to specifically indicate the three regions of the protein in the amino acid sequence, and the figure caption now provides more detailed information. 

  1. L185

Isn't "Supplementary Materials" "Figure 3"? The description here does not match the description in Supplementary Materials. Line 183

Response:

The observation is adequate, we have added a figure in supplementary material showing the structure of the GroEL proteins.

  1. L214

There is no instruction for “Figure 4” in the text. It would probably be appropriate to add (Figure 4) to the end of L214.

Response:

We appreciate the observation, we have added "Figure 4" to the text, lines 211 and 215.

  1. Figure 4

The graph in Figure 4 is difficult to understand. The legend to the right of the graph should be removed and the names of all the samples in the graph should be written on the X-axis.

Response:

Thank you for this comment. As suggested by the reviewer, the figure and text have been modified and is now clearer to understand.

  1. L221 P>0.0001→ P 0.0001

L235   GroELPI → GroELPl

L246,   L247 cm2 → cm2

L310    pH: 7.3 → pH 7.3

L314    Mg2+   → Mg2+

Response:

Thank you for these observations. The suggested correction has been made.

The authors want to thank the comments of the reviewer that improve substantially the manuscript.

Reviewer 2 Report

Article about chapertonin GroEL and its toxic effect on insects. The authors evaluated the chaperone and insecticidal activities of various GroEL proteins from entomopathogenic nematodes on G. mellonella. The ability to synergize with Pseudomonas aeruginosa exotoxin A protein was also tested. The GroELXn protein showed the greatest insecticidal activity, and by activating the phenoloxidase system, we can talk about the mechanism of toxic effects on insects. Materials and methods written in an understandable way. 5.8 Statistical Analysis it is not written what tests were performed. A small number of literature items. Figures simple but clear.

Author Response

We appreciate the positive feedback from the reviewer. We have answered each of your points below.

  1. Statistical Analysis it is not written what tests were performed.

Response:

The observation is adequate, we have added in section 5 Subject and Methods, the statistical analyzes used. Lines 300-302, 336-338 and 347–349.

  1. A small number of literature items.

 Response:  

Thank you for raising this point. We conducted a comprehensive review of the manuscript and have included new bibliographic citations that provide clearer support for the writing.

The authors want to thank the comments of the reviewer that improve substantially the manuscript.